# Adding salt to foods increases the risk of metabolic dysfunction-associated steatotic liver disease
Han Chen[1,2], Xujun Zhang[3], Shujuan Lin [ ][4] ✉ & Qiong Wu [ ][1] ✉

## Abstract

**Background:** Although salt intake has been linked to multiple cardiometabolic diseases, whether the frequency of adding salt to foods, a reasonable proxy for long-term salt intake, is related to metabolic dysfunction-associated steatotic liver disease (MASLD) incidence remains unknown.
**Methods:** This prospective study included 494,110 UK Biobank participants (mean age 56.5 years) who were free of MASLD at baseline. Participants were followed for a median of 13.6 years. Cox proportional hazards models were used to examine the relationship between the frequency of adding salt to foods and incident MASLD. Mediation analyses explored the role of blood biomarkers, and interaction analyses assessed whether genetic factors modify this association.
**Results:** Here, we show that among the cohort, 7171 participants develop MASLD during follow-up. Compared to people who never or rarely adding salt, those who sometimes, usually, and always add salt to foods have 7%, 20%, and 35% higher risk, respectively. This association is stronger in people with normal body mass index and those who frequently drink alcohol. Blood markers of inflammation and metabolism, such as C-reactive protein, insulin-like growth factor-1, triglycerides, and urate, partially mediate this relationship. A significant interaction is observed, with *PNPLA3* genetic susceptibility amplifying the MASLD risk associated with frequently adding salt to foods.
**Conclusions:** A higher frequency of adding salt to foods is associated with increased MASLD risk. Reducing table salt use represents a simple, actionable strategy for disease prevention, particularly for genetically susceptible individuals.

## Plain language summary

Many people add salt to their food at the table, but it is not clear whether this habit affects liver health. We studied nearly 500,000 British adults for about 14 years to find out if people who frequently add salt to their meals are more likely to develop fatty liver disease, a condition in which fat builds up in the liver, which can cause serious health problems. We find that people who always add salt to their food have a 35% higher chance of developing fatty liver disease compared to those who rarely or never add salt. This risk is especially high in people with certain inherited traits, those who regularly drink alcohol, and people of normal weight. Our findings suggest that reducing salt added at the table could help protect your liver and this should be recommended to people to reduce the risk of developing liver disease.

---

Metabolic dysfunction-associated steatotic liver disease (MASLD) has recently been proposed to replace the previously used term non-alcoholic fatty liver disease (NAFLD). It is characterized by excessive fat accumulation in the liver, accompanied by at least one criterion of metabolic dysfunction[1]. MASLD is one of the most prevalent chronic liver diseases worldwide, with an estimated global prevalence of around 32%[2]. Given its close association with other highly prevalent chronic conditions like obesity and insulin resistance, the prevalence of MASLD projected to rise even further[3-5]. Therefore, identifying modifiable risk factors is essential for recognizing

individuals at high risk and developing effective prevention and treatment strategies.

A high-salt diet has emerged as a significant dietary risk factor for chronic non-communicable diseases, particularly cardio-metabolic conditions such as hypertension and type 2 diabetes, both of which are established risk factors for MASLD[6]. Animal studies suggest that excessive salt intake contributes to the onset and progression of MASLD and its related metabolic disorders[7,8]. In mice, high salt consumption has been shown to exacerbate hepatic steatosis, inflammation, and fibrosis, likely through

---

[1]Department of Epidemiology and Biostatistics, School of Public Health and Nursing, Hangzhou Normal University, Hangzhou, China. [2]Department of Big Data in Health Science, School of Public Health, Zhejiang University School of Medicine, Hangzhou, China. [3]Department of Pathology and Pathophysiology, School of Basic Medical Sciences, Hangzhou Normal University, Hangzhou, China. [4]School of Basic Medicine Science, Key Laboratory of Translational Tumor Medicine in Fujian Province, Putian University, Putian, China. ✉e-mail: sjlin@zju.edu.cn; 20220094@hznu.edu.cn

mechanisms involving oxidative stress, insulin resistance, and lipid metabolism dysregulation[7,9,10]. These findings underscore the potential preventive effect of salt reduction on MASLD.

However, large-scale epidemiological studies concerning the relationship between dietary salt intake and MASLD risk are scarce. Most existing studies rely on cross-sectional designs, which are unable to establish temporal relationships[11–16]. Furthermore, previous studies typically estimated dietary sodium intake based on a single day's urine collection or 24-hour dietary recall survey, which is insufficient to capture habitual consumption levels of sodium due to day-to-day variability[17]. Alternatively, the frequency of adding salt to foods (usually at table), a common eating behavior in Western countries, was highly correlated with the preference for food saltiness and the willingness to consume salty foods[18,19]. As such, the frequency of adding salt to foods could serve as a reasonable proxy for long-term salt intake[20]. To date, no study has assessed whether the frequency of adding salt to foods is associated with MASLD risk in a prospective setting. Recently, a Mendelian randomization study provided genetic evidence supporting a potential causal relationship between genetically-proxied salt preference and MASLD risk. While these findings enhance the biological plausibility of the salt-MASLD link, prospective population-based studies assessing dietary behaviors remain limited and are needed to validate this association in real-world settings[21]. Furthermore, while preclinical and experimental studies have proposed mechanisms underlying the salt intake-MASLD link, the mediating role of specific blood biomarkers, indicating inflammation and pathophysiological alterations, remains unverified in population-based cohorts[22].

In addition to environmental factors, genetics also contribute to MASLD risk[23]. Genome-wide association studies have identified many genetic variants associated with an increased susceptibility to MASLD[24,25]. Research on gene-environment interactions emphasized the critical importance of maintaining a healthy diet to reduce the genetic predisposition to MASLD[26]. Thus, investigating the interactive effects of genetic predisposition and dietary salt intake on MASLD risk is crucial.

In this large-scale longitudinal study using data from the UK Biobank, we find that a higher frequency of adding salt to foods is associated with increased MASLD risk in a dose-dependent manner. This association is partially mediated by metabolic and inflammatory biomarkers, including insulin-like growth factor-1, C-reactive protein, triglycerides, and urate. Among individuals with existing MASLD, frequent salt addition is associated with a threefold higher risk of advanced liver fibrosis. Furthermore, we identify a significant gene-environment interaction whereby individuals carrying the *PNPLA3* risk genotype who frequently add salt to foods exhibit the highest MASLD risk. These findings suggest that reducing table salt use represents a simple, modifiable dietary intervention for MASLD prevention, particularly for genetically susceptible individuals.

## Methods
### Study population
The UK Biobank is a large population-based prospective cohort study that recruited over 500,000 participants, aged between 37 and 73 years, across 22 assessment centers in England, Scotland, and Wales between 2006 and 2010. During the baseline assessment, participants completed nurse-administered touchscreen questionnaires covering demographic characteristics and lifestyle and health-related information, underwent physical examinations, and provided biological samples for genotyping and laboratory assay. Written informed consent was obtained from all participants, and the study was approved by the North West Multi-Center Research Ethics Committee (R21/NW/0157). This approval covers our research purposes, so no additional ethical approval was required. Details of the study design and data collection have been described previously [27]. Additionally, approximately 40,000 participants were invited to undergo abdominal magnetic resonance imaging (MRI) as part of the UK Biobank's imaging enhancement project[28]. The study was conducted using the UK Biobank data under application number 95817.

In this study, using data from the UK Biobank, we conducted a large-scale longitudinal study to investigate the association between the frequency of adding salt to foods and MASLD risk. Additionally, we explored potential mediating effects of blood biomarkers on these associations and assessed the joint effect of adding salt to foods and genetic predictors on MASLD risk. The design and workflow of the present study are illustrated in Fig.1 and Supplementary Fig.1. From the initial sample of 502,364 participants, we excluded 1708 participants with incomplete data on the frequency of adding salt to foods, and those with prevalent MASLD at baseline. Following the latest Expert Panel Consensus Statement[29], an additional 6546 participants with other liver diseases or alcohol/drug use disorder at baseline were excluded. The diagnostic codes for these exclusions are detailed in Supplementary Table 1. This left 494,110 participants for the primary analysis. In a subset of 473,110 participants, the association between estimated 24-hour urinary sodium excretion and MASLD incidence was further explored, after excluding 20,982 individuals with missing data necessary for calculating estimated sodium excretion and 28 individuals with negative excretion values. Furthermore, in the primary dataset, 40,257 participants who underwent liver MRI scans were included to investigate the relationship between the frequency of adding salt to foods and liver fat content. Blood biomarker data, available for 40,443 to 462,232 participants, were used to assess the mediating effects of these biomarkers on the association between the frequency of adding salt to foods and MASLD risk. Genotype data from 479,589 participants were analyzed to explore the joint effect of genetic predisposition and the frequency of adding salt to foods on MASLD risk.

### Exposure assessment
During the baseline visit, participants were instructed to complete a food frequency questionnaire, which included the question: "Do you add salt to your foods? (Do not include salt used in cooking)." Participants could choose from one of five response options: 1) never/rarely; 2) sometimes; 3) usually; 4) always; or 5) prefer not to answer. Those who selected "preferred not to answer" were considered to have missing information and were excluded from the analysis.

Urinary sodium, potassium, and creatinine were analyzed from a random urine sample collected at baseline using a single Beckman Colter AU5400[30]. Estimated 24 h urinary sodium excretion was calculated from the spot urinary concentration values using the sex-specific INTERSALT (International Cooperative Study on Salt, Other Factors, and Blood Pressure) equation[31,32].

### Outcome assessment
MASLD was identified through record linkage to electronic health records (England and Wales: Health Episode Statistics; Scotland: Scottish Morbidity Records). Detailed information about the linkage procedures is available at http://content.digital.nhs.uk/services. It should be noted that while we use the term MASLD throughout this manuscript to align with current nomenclature, case identification was based on the previous NAFLD criteria, as the International Classification of Diseases-10th Revision (ICD-10) coding system predates the recent transition from NAFLD to MASLD terminology. According to the Expert Panel Consensus Statement[29], incident MASLD was defined using ICD-10 code K76.0 for nonalcoholic fatty liver disease and K75.8 for nonalcoholic steatohepatitis. Participants were followed until the earliest occurrence of MASLD, death, or the end of the follow-up period (September 31, 2022), whichever came first.

Additionally, for participants who underwent liver MRI, liver fat content was quantified using the proton density fat fraction (PDFF), which represents the percentage of fat within the liver[33]. According to previous literature, MASLD was defined as a PDFF > 5.5%[34].

### Assessment of MASLD severity
We calculated the Fibrosis-4 (FIB-4) index based on age, aspartate aminotransferase (AST), alanine aminotransferase (ALT), and platelet count among participants with MASLD at baseline[35]. The FIB-4 index was

# Adding salt to foods and risk of MASLD

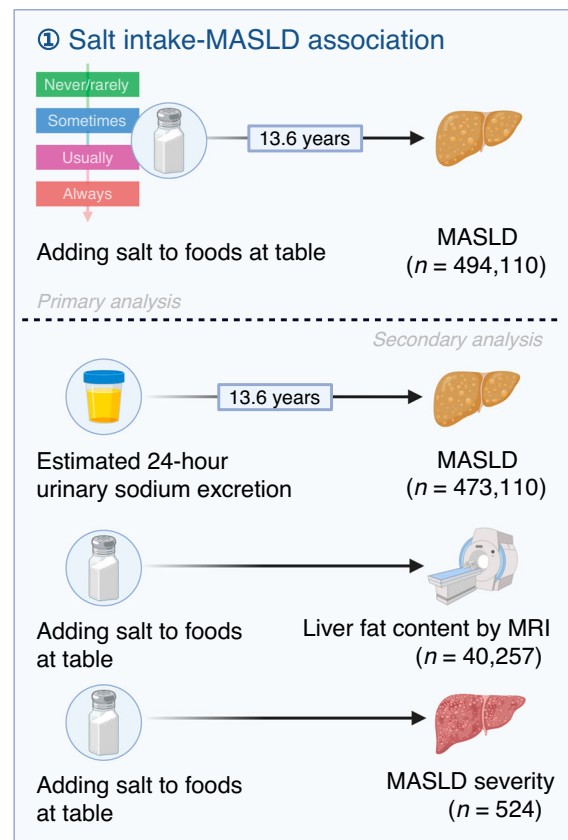

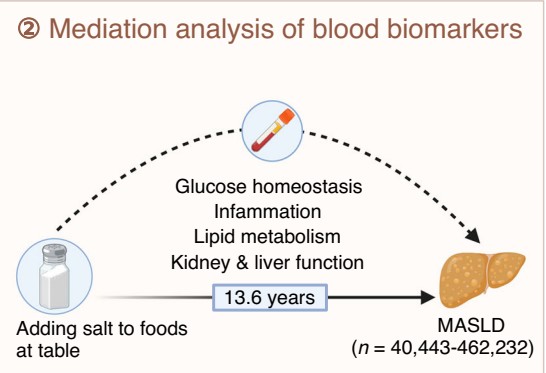

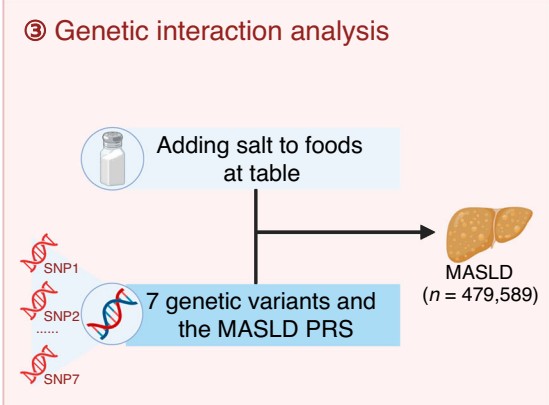

**Fig. 1 | Overview of analyses performed in the current study.** Leveraging data from the UK Biobank, this study investigates the prospective association of the frequency of adding salt to foods and risk of incident MASLD. This study subsequently conducts secondary analyses using the estimated 24-hour urinary sodium excretion as a marker of sodium consumption, using proton density fat fraction to quantify liver fat content by MRI techniques, and using Fibrosis-4 index to access MASLD severity. Then, this study examines the longitudinal mediation effect of blood biomarkers on the association between the frequency of adding salt to foods and MASLD risk. Finally, this study examines the joint interaction effect of genetic predictors and the frequency of adding salt to foods on MASLD risk. Created with https://BioRender.com. Abbreviation: *MASLD* metabolic dysfunction-associated steatotic liver disease, *MRI* magnetic resonance imaging, *PRS* polygenic risk score, *SNP* single nucleotide polymorphism.

estimated as follows (age in years, ALT and AST in IU/L, and platelet count in $10^9$/L):

$$\text{FIB-4 index} = \frac{(\text{age} \times \text{AST})}{\text{platelet count} \times \sqrt{\text{ALT}}} \quad (1)$$

Following established criteria, we classified participants with FIB-4 > 2.67 as having a high risk of advanced fibrosis, and those with FIB-4 ≤ 2.67 as having a low/indeterminate risk of liver fibrosis[36,37].

## Covariates

Potential confounding variables were selected based on *a priori*-defined directed acyclic graph (Supplementary Fig. 2), including age at baseline assessment, sex (female or male), ethnicity (non-White or White), Townsend deprivation index, education level (less than high school, high school or equivalent, and college or above), healthy diet score, smoking status (never, previous, and current), alcohol consumption (none currently, up to twice per week, and three or more times per week), physical activity level (low, moderate, and high), metabolic syndrome severity, cancer (no or yes), and cardiovascular disease (no or yes). The Townsend deprivation index reflects the level of socioeconomic deprivation on participants' post codes. It was derived from aggregated data of unemployment, car ownership, house ownership, and household overcrowding, with higher scores indicating higher deprivation[38]. The healthy diet score (ranging from 0–7) was constructed by assigning 1 point for a healthy frequency and 0 point for an unhealthy frequency of consuming fruits, vegetables, fish, processed meat, unprocessed red meat, whole grains, and refined grains, with higher scores indicating a healthier diet quality[39]. Metabolic syndrome severity was defined using baseline data (ranging from 0–5), including central obesity (waist circumstance ≥ 102 cm in men and ≥ 88 cm in women), high glycaemia (fasting glucose ≥ 6.1 mmol/L), high blood pressure (systolic blood pressure ≥ 130 mmHg, diastolic blood pressure ≥ 85 mmHg, or a prior diagnosis of hypertension), low high-density lipoproteins cholesterol (HDL-C) (≤ 1.0 mmol/L in men and ≤ 1.3 mmol/L in women), and high triglycerides (≥ 1.7 mmol/L)[40]. Bodyweight and height were measured at baseline, with body mass index (BMI) calculated as weight (kg)/(height (m)$^2$). Further details on covariates assessment are described in the Supplementary Table 2.

## Mediators

Thirty blood biomarkers were included in this study, measured from blood samples collected at baseline (UK Biobank category 17518). These features fall into five main categories: (1) glucose homeostasis ($n = 3$; i.e., hemoglobin A1c [HbA1c], glucose, and insulin-like growth factor-1 [IGF-1]); (2) inflammation ($n = 2$; i.e., C-reactive protein [CRP], and rheumatoid factor); (3) kidney function ($n = 7$; e.g., urate, creatinine, and cystatin C); (4) lipid metabolism ($n = 7$; e.g., HDL-C, total cholesterol, and triglycerides); (5) liver function ($n = 11$; e.g., alkaline phosphatase [ALP], gamma glutamyl-transferase [GGT], and sex hormone-binding globulin [SHBG]). Detailed

information regarding blood biomarkers is presented in Supplementary Table 3. The procedures for blood sample collection have previously been described and validated[41]. Data were standardized using z-score prior to analyses.

## Genotype data

Genotyping was conducted using one of two custom arrays: the UK BiLEVE Axiom Array or UK Biobank Axiom Array. Imputation was performed using the Haplotype Reference Consortium and UK10K reference panel. Details regarding genotyping, imputation, and quality control of genetic data in UK Biobank has been discussed elsewhere[42]. This study included seven specific genetic variants (single nucleotide polymorphism, SNP) that are significantly associated with MASLD: *PNPLA3*-rs738409, *TM6SF2*-rs58542926, *HSD17B13*-rs72613567, *MTARC1*-rs2642438, *APOE*-rs429358, *GCKR*-rs1260326, and *GPAM*-rs2792751[24,25]. Additionally, the MASLD polygenic risk score (PRS) was calculated based on the dosages of these individual genetic variants, each weighted by its corresponding effect estimate, and subsequently standardized using z-scores[24].

## Statistics and reproducibility

Baseline characteristics by the frequency of adding salt to foods were described as means and standard deviations (SDs) for continuous variables, and as numbers (percentages) for categorical variables. Comparisons across different frequencies of adding salt to foods were performed using one-way ANOVA for continuous variables and Pearson's Chi-square tests for categorical variables. Associations between the frequency of adding salt to foods (using never/rarely adding salt to foods as the reference), quartiles of estimated 24-hour urinary sodium excretion (with the lowest quartile as the reference), and MASLD risk were examined using multivariable Cox proportional hazard models. Results are reported as hazard ratios (HRs) and 95% confidence intervals (CIs). The proportional hazards assumption was tested by Schoenfeld residuals and no violations were observed. The associations between the frequency of adding salt to foods and PDFF and PDFF-defined MASLD was evaluated using multivariable linear regression and logistic regression, respectively. The associations between the frequency of adding salt to foods and continuous FIB-4 index and binary FIB-4 index were evaluated using multivariable linear regression and logistic regression, respectively. Results are reported as regression coefficients (β), odds ratio (OR), and corresponding 95% CIs. Model 1 adjusted for age and sex. Model 2 further adjusted for ethnicity, Townsend deprivation index, education level, healthy diet score, smoking status, alcohol consumption, physical activity level, metabolic syndrome severity, cancer, and cardiovascular disease. Missing data for covariates were imputed using multiple imputation by chained equations, with five imputations[43]. Spearman correlations were used to access the consistency across multiple assessments of frequency of adding salt to foods.

The longitudinal mediation effect of thirty blood biomarkers on the association between the frequency of adding salt to foods and MASLD was assessed using *mediate* macro in SAS software (https://ysph.yale.edu/cmips/research/software/analysis-graphics/mediate/). The *mediate* SAS macro compares a full model, which includes the exposure, a hypothesized mediator, and any covariates, with a partial model that excludes the mediator, calculating the proportion of mediation[44]. In this study, the mediation model was adjusted for age, sex, ethnicity, Townsend deprivation index, education level, healthy diet score, smoking status, alcohol consumption, physical activity level, metabolic syndrome severity, cancer, and cardiovascular disease.

To investigate the joint interaction effect of genetic predictors and the frequency of adding salt to foods on MASLD risk, participants were classified into twelve groups based on both the frequency of adding salt to foods (never/rarely, sometimes, usually, and always) and individual SNP genotype (e.g., C/C, C/G, and G/G genotypes) in the fully adjusted Cox models. The reference group consisted of participants who answered "never/rarely adding salt to foods" and carried the lowest genetic risk of MASLD.

A series of sensitivity analyses was conducted to test the robustness of the results. First, participants with missing values in covariates were excluded to account for missing data. Second, to minimize the potential reverse causation, participants with metabolic syndrome, hypertension or diabetes at/before baseline were excluded. Third, to further address reverse causation, participants who developed MASLD within the first two years of follow-up were also excluded. Fourth, to reduce selection bias, we excluded participants who had made dietary changes in the past 5 years due to illness or other reasons. Participants were asked, "Have you made any major changes to your diet in the last five years?" and chose one of the following options: "No," "Yes, because of illness," "Yes, because of other reasons," and "Prefer not to answer" during the baseline assessment. Fifth, we included MASLD cases identified by using primary care, hospitalizations and death records. Sixth, we further adjusted for sugar-sweetened beverages, saturated fatty acids, free sugar, fiber, and total energy from 24 h dietary recalls[20,45]. Finally, we included an additional adjustment for BMI.

We also conducted stratified analyses by age ( < 60 years or ≥60 years), sex (female or male), ethnicity (non-White or White), Townsend deprivation index ( < median, ≥median), education level (no qualification, any other qualification, and higher education), healthy diet score ( < 4 or ≥4), smoking status (never, previous, and current), alcohol consumption (none currently, up to twice per week, and three or more times per week), physical activity level (low, moderate, and high), BMI (18.5–24.9 kg/m$^2$, 25–29.9 kg/m$^2$, and ≥30 kg/m$^2$), metabolic syndrome severity ( < 3 or ≥3), cancer (no or yes), and cardiovascular disease (no or yes). Multiplicative interactions were evaluated via a likelihood ratio test comparing modes with and without cross-product terms.

Mediation analyses were performed using SAS version 9.4 (SAS Institute), and the main analyses were conducted using R version 4.3.1. Statistical significance was defined as two-sided $P < 0.05$.

## Reporting summary

Further information on research design is available in the Nature Portfolio Reporting Summary linked to this article.

## Results

### Population characteristics

During a median follow-up time of 13.6 years (interquartile range 12.2–15.1 years), 7171 (1.45 %) cases of incident MASLD were recorded. Baseline characteristics of participants, categorized by the frequency of adding salt to foods, are presented in Table 1. Compared to participants with a lower frequency of adding salt to foods, those with a higher frequency were more likely to be male, to be non-White, and have a higher Townsend deprivation index and a lower level of education; These participants also tended to have a lower healthy diet score, were more likely to be current smokers, less physically active, and exhibit a higher severity of metabolic syndrome. Additionally, they had a higher prevalence of cancer and cardiovascular disease. Baseline characteristics of participants with available estimated 24-hour urinary sodium excretion and participants with liver MRI data are presented in Supplementary Table 4.

### Association between the frequency of adding salt to foods and MASLD risk

Table 2 outlines the association between the frequency of adding salt to foods and the risk of incident MASLD. In the fully adjusted model (Model 2), a higher frequency of adding salt to foods was significantly associated with an increased risk of MASLD. Compared to the reference group, the multivariable-adjusted HRs and 95% CIs were 1.07 (1.02–1.13), 1.20 (1.11–1.28), and 1.35 (1.23–1.48) for participants who sometimes, usually, and always adding salt to foods, respectively ($P$ for trend = $3.66 \times 10^{-13}$). Results were consistent across several sensitivity analyses (Supplementary Table 5).

To complement these findings, we conducted a secondary analysis using the estimated 24-hour urinary sodium excretion as a marker of sodium consumption. The mean concentrations of estimated 24-hour

**Table 1 | Baseline characteristics according to the frequency of adding salt to foods**

| Characteristic | Participants (N = 494,110) | | | | |
|---|---|---|---|---|---|
| | Never/rarely(n = 274,781) | Sometimes(n = 138,627) | Usually(n = 57,190) | Always(n = 23,512) | P value |
| Age, years | 56.5 (8.1) | 56.4 (8.1) | 57.0 (8.0) | 56.0 (8.3) | $6.71 \times 10^{-74}$ |
| Sex | | | | | $2.59 \times 10^{-222}$ |
| Female | 154,686 (56.3%) | 75,256 (54.3%) | 28,173 (49.3%) | 12,305 (52.3%) | |
| Male | 120,095 (43.7%) | 63,371 (45.7%) | 29,017 (50.7%) | 11,207 (47.7%) | |
| Ethnicity | | | | | $1 \times 10^{-350}$ |
| Non-White | 11,505 (4.2%) | 8794 (6.3%) | 3599 (6.3%) | 2871 (12.2%) | |
| White | 263,276 (95.8%) | 129,833 (93.7%) | 53,591 (93.7%) | 20,641 (87.8%) | |
| Townsend deprivation index | −1.5 (3.0) | −1.2 (3.1) | −1.1 (3.2) | −0.3 (3.5) | $1 \times 10^{-350}$ |
| Education level | | | | | $1 \times 10^{-350}$ |
| Less than high school | 41,960 (15.3%) | 24,907 (18.0%) | 11,002 (19.2%) | 7,088 (30.1%) | |
| High school or equivalent | 136,863 (49.8%) | 69,673 (50.3%) | 28,699 (50.2%) | 11,474 (48.8%) | |
| College or above | 95,958 (34.9%) | 44,047 (31.8%) | 17,489 (30.6%) | 4950 (21.1%) | |
| Healthy diet score | 3.8 (1.5) | 3.5 (1.5) | 3.3 (1.5) | 3.0 (1.5) | $1 \times 10^{-350}$ |
| Smoking status | | | | | $1 \times 10^{-350}$ |
| Never | 163,501 (59.5%) | 73,045 (52.7%) | 26,052 (45.6%) | 9666 (41.1%) | |
| Previous | 89,673 (32.6%) | 50,186 (36.2%) | 22,660 (39.6%) | 8558 (36.4%) | |
| Current | 21,607 (7.9%) | 15,396 (11.1%) | 8478 (14.8%) | 5288 (22.5%) | |
| Alcohol consumption | | | | | $1 \times 10^{-350}$ |
| None currently | 22,212 (8.1%) | 10,047 (7.2%) | 4274 (7.5%) | 2914 (12.4%) | |
| Up to twice per week | 137,675 (50.1%) | 66,942 (48.3%) | 25,366 (44.4%) | 10,680 (45.4%) | |
| Three or more times per week | 114,894 (41.8%) | 61,638 (44.5%) | 27,550 (48.2%) | 9918 (42.2%) | |
| Physical activity level | | | | | $8.18 \times 10^{-100}$ |
| Low | 49,106 (17.9%) | 26,565 (19.2%) | 11,640 (20.4%) | 5244 (22.3%) | |
| Moderate | 112,780 (41.0%) | 56,066 (40.4%) | 22,734 (39.8%) | 8743 (37.2%) | |
| High | 112,895 (41.1%) | 55,996 (40.4%) | 22,816 (39.9%) | 9525 (40.5%) | |
| Metabolic syndrome severity | 2.1 (1.3) | 2.2 (1.3) | 2.2 (1.3) | 2.3 (1.3) | $2.98 \times 10^{-166}$ |
| Cancer | 22,885 (8.3) | 11,560 (8.3) | 5002 (8.7) | 1995 (8.5) | 0.009 |
| Cardiovascular disease | 43,407 (15.8) | 21,601 (15.6) | 9022 (15.8) | 4041 (17.2) | $1.76 \times 10^{-8}$ |

Data are n (%) or mean (SD). P values were calculated based on the one-way ANOVA for continuous variables, and Pearson's Chi-square tests for categorical variables.

**Table 2 | Association between the frequency of adding salt to foods and risk of MASLD**

| | Frequency of adding salt to foods, HR (95% CI) | | | | P for trend |
|---|---|---|---|---|---|
| | Never/rarely | Sometimes | Usually | Always | |
| Events No./ total No. | 3576/274781 | 2073/138627 | 985/57190 | 537/23512 | |
| Model 1 | 1 (reference) | 1.15 (1.09–1.22) | 1.32 (1.23–1.42) | 1.81 (1.65–1.98) | $1.52 \times 10^{-41}$ |
| Model 2 | 1 (reference) | 1.07 (1.02–1.13) | 1.20 (1.11–1.28) | 1.35 (1.23–1.48) | $3.66 \times 10^{-13}$ |

Model 1: Adjusted for age, sex.
Model 2: Model 1 + ethnicity, Townsend deprivation index, education level, healthy diet score, smoking status, alcohol consumption, physical activity level, metabolic syndrome severity, cancer, and cardiovascular disease.
Abbreviation: *HR* hazard ratio, *CI* confidence interval.

urinary sodium excretion were 2.92 g (SD: 0.77 g), 3.01 g (SD: 0.81 g), 3.12 g (SD: 0.84 g), and 3.15 g (SD: 0.85 g), in the never/rarely, sometimes, usually, and always adding salt to foods groups, respectively. Supplementary Table 6 presents the association between estimated 24-hour urinary sodium excretion and MASLD risk. In Model 2, higher concentrations of estimated 24-hour urinary sodium excretion was significantly associated with an increased risk of MASLD. Compared to participants with the lowest quartile of estimated 24-hour urinary sodium excretion (0 to 2.37 g), the adjusted HR (95% CI) were 1.36 (1.25–1.47), 1.89 (1.73–2.06), and 2.61 (2.36–2.88)

across quartile 2 (2.37 to 2.86 g), quartile 3 (2.86 to 3.53 g), and quartile 4 (3.53 to 9.08 g), respectively ($P$ for trend = $2.05 \times 10^{-104}$).

In the other secondary analysis, we further examined the association of the frequency of adding salt to foods with PDFF, and PDFF-defined MASLD. The results are presented in Supplementary Table 7. In Model 2, a higher frequency of adding salt to foods was significantly associated with a higher PDFF percentage. Compared to the reference group, the adjusted β (95% CI) were 0.31 (0.21–0.42), 0.32 (0.17–0.47), 0.31 (0.04–0.58) for the groups that sometimes, usually, and always adding salt to foods, respectively

(P for trend = $8.22 \times 10^{-9}$). Similarly, a higher frequency of adding salt to foods was significantly associated with an increased risk of PDFF-defined MASLD. The adjusted OR and 95% CI were 1.17 (1.10–1.23), 1.18 (1.09–1.28), 1.19 (1.03–1.36) across the groups of sometimes, usually, and always adding salt to foods, respectively (P for trend = $3.59 \times 10^{-8}$).

### Association between the frequency of adding salt to foods and MASLD severity

To evaluate the association between the frequency of adding salt to foods and MASLD severity, we calculated the FIB-4 index among participants with MASLD at baseline[35]. Among 524 participants with MASLD at baseline who had complete data for FIB-4 index calculation, we examined the association between the frequency of adding salt to foods and liver fibrosis risk. In the fully adjusted model, compared to those who never/rarely added salt to foods, participants who always added salt showed a trend toward higher FIB-4 scores (β = 0.37, 95% CI: 0.03–0.70), although this did not reach statistical significance for the continuous outcome. However, when analyzed as a binary outcome, participants who always added salt to foods demonstrated a significantly higher risk of advanced fibrosis (FIB-4 > 2.67) compared to the reference group (OR = 3.55, 95% CI: 1.36–8.85) (Supplementary Table 8).

### Stratified analyses

Stratified analyses were conducted to assess whether the covariates modified the association of the frequency of adding salt to foods with MASLD (Supplementary Data 1). A higher frequency of adding salt to foods was significantly associated with a higher risk of MASLD across different groups, including age, sex, smoking status, and across different levels of Townsend deprivation index, education, healthy diet score, physical activity, metabolic syndrome severity, and BMI (all P for trends <0.05). The significant positive associations were also observed among participants with and without baseline cardiovascular disease. However, the association was significant for White participants but not for non-White participants, and for current alcohol consumers but not for non-current alcohol consumers. Similarly, non-cancer patients had a significant association between the frequency of adding salt to foods and MASLD risk, while cancer patients did not.

In the interaction evaluation, significant interactions were identified between BMI, alcohol consumption, and the frequency of adding salt to foods on the risk of MASLD (P values for interaction <0.05). Participants who consumed alcohol three or more times per week exhibited the strongest association between the frequency of adding salt to foods and MASLD risk (adjusted HR [95% CI]: 1 [reference], 1.16 [1.06–1.28], 1.30 [1.16–1.45], 1.49 [1.27–1.74] for never/rarely, sometimes, usually, always adding salt to foods, respectively; P for trend = $1.41 \times 10^{-10}$). Additionally, the positive association between the frequency of adding salt to foods and MASLD was more pronounced in individuals with BMI < 25 kg/m$^2$ (adjusted HR [95% CI]: 1 [reference], 1.13 [0.95–1.35], 1.19 [0.94–1.51], 2.02 [1.55–2.62] for never/rarely, sometimes, usually, always adding salt to foods, respectively; P for trend = $3.48 \times 10^{-6}$) (Supplementary Data 1).

### Mediation effects of blood biomarkers

Significant longitudinal mediation effects (P values for mediation effect <0.05) of blood biomarkers on the association between the frequency of adding salt to foods and MASLD are shown in Fig.2. For the associations between sometimes vs. never/rarely adding salt to foods and the MASLD risk, CRP, HbA1c, IGF-1, SHBG, triglycerides, and urate significantly mediated the relationship, with mediation proportions of 7.3%, 3.8%, 28.8%, 2.0%, 5.2%, and 8.3%, respectively. In the comparison of usually vs. never/rarely adding salt to foods, CRP, IGF-1, triglycerides, and urate were significant mediators, with mediation proportions of 3.5%, 18.8%, 4.5%, and 2.0%, respectively. For the always vs. never/rarely adding salt to foods group, ALP, CRP, GGT, HbA1c, IGF-1, triglycerides and urate played significant mediating roles, with mediation proportions of 3.5%, 4.7%, 17.4%, 4.8%, 17.5%, 2.8%, and 2.3%, respectively. Detailed results regarding the mediating effect of all the 30 blood biomarkers are provided in Supplementary Data 2.

### The joint effect of the frequency of adding salt to foods and genetic predictors on MASLD risk

Participants were classified according to the joint categories of the frequency of adding salt to foods and SNP genotypes/tertiles of the MASLD PRS, with the never/rarely adding salt to foods group and the lowest genetic risk of MASLD serving as the reference. A significant interaction was observed between the frequency of adding salt to foods and PNPLA3-rs738409. Compared with the reference group, participants who always added salt to foods and carried two risk alleles for PNPLA3-rs738409 (G/G genotype) had the highest risk of MASLD (adjusted HR [95% CI]: 3.01 [2.26-4.00]; P for interaction = 0.025).

For APOE-rs429358, HSD17B13-rs72613567, MTARC1-rs2642438, GCKR-rs1260326, participants who always added salt to foods and carried two risk alleles for each SNP genotype exhibited the highest risk of MASLD, with adjusted HR (95% CI) of 2.33 (1.75–3.11), 1.40 (1.01–1.93), 1.73 (1.46–2.07), and 1.78 (1.44–2.19), respectively. However, tests for interaction effect were not significant (P for interaction = 0.779, 0.780, 0.832, and 0.515, respectively). A nominal interaction effect was observed for the frequency of adding salt to foods and the MASLD PRS (P for interaction = 0.098), with participants who always added salt to foods and were in the highest MASLD PRS tertile (T3) exhibiting the highest risk of MASLD, with adjusted HR (95% CI) of 2.39 (2.07–2.77) (Fig.3 and Supplementary Data 3).

### Discussion

In this large prospective cohort study, we found that a higher frequency of adding salt to foods was significantly associated with an increased risk of developing MASLD, Both the secondary analyses and sensitivity analyses confirmed the consistency of these associations. Stratified analyses revealed that the association between frequency of adding salt to foods and MASLD was more pronounced among participants with a normal BMI, and current alcohol consumer participants. Furthermore, blood biomarkers, particularly IGF-1, CRP, triglycerides, and urate, were found to partially mediate this association. Additionally, we identified a significant interaction between the frequency of adding salt to foods and the PNPLA3-rs738409 genotype in relation to MASLD risk. Participants with G/G genotype for PNPLA3-rs738409 exhibited an amplified adverse effect of salt intake on the development of MASLD.

Previous studies have primarily focused on total salt intake or urinary sodium excretion as markers of salt consumption. A cross-sectional study of 6132 participants from the Prevention of Renal and Vascular End-Stage Disease (PREVEND) cohort in the Netherlands found that higher 24 h urinary sodium excretion was associated with increased prevalence of MASLD[12]. Similarly, another cross-sectional study of 11,022 participants from the U.S. reported a positive association between dietary sodium intake and MASLD prevalence. While 24 h urinary sodium excretion is considered the gold standard for assessing salt intake, it is subject to day-to-day variability and may not capture long-term habitual intake[46]. In contrast, the frequency of adding salt to foods represents a specific, modifiable dietary behavior that contributes significantly to overall salt intake and may serve as a surrogate marker of long-term salt preference and consumption. Previous studies have validated the frequency of adding salt to foods against objectively measured urinary sodium concentrations in a UK population[20,47]. Our study contributes to the existing evidence by highlighting a readily modifiable aspect of salt consumption. The consistent associations observed across different measures of salt intake (self-reported frequency of adding salt to foods, and estimated 24 h urinary sodium excretion) and MASLD definition (ICD-based MASLD, and PDFF-defined MASLD) further strengthen the robustness of our findings. In addition, a Mendelian randomization study using multiple genome-wide association study datasets identified salt added to food as a potential causal risk factor for MASLD, providing genetic support for our findings[21]. Building upon this, our study contributes complementary observational evidence by evaluating salt-adding behavior in a large prospective cohort, accounting for multiple potential confounders and conducting a series of secondary analyses.

**a**  Sometimes *vs.* Never/rarely to MASLD

| Mediators | HR (95% CI) without adjusting for mediator | HR (95% CI) adjusting for mediator | Proportion explained by mediator (%) | | *P* for mediation |
|---|---|---|---|---|---|
| CRP | 1.07 (1.01-1.14) | 1.07 (1.01-1.13) | | 7.3 | 6.02×10⁻⁶ |
| HbA1c | 1.06 (1.00-1.12) | 1.06 (1.00-1.12) | | 3.8 | 0.004 |
| IGF–1 | 1.08 (1.02-1.14) | 1.05 (1.00-1.12) | | 28.8 | 1×10⁻³⁵⁰ |
| SHBG | 1.07 (1.01-1.14) | 1.07 (1.01-1.14) | | 2.0 | 0.019 |
| Triglycerides | 1.07 (1.01-1.14) | 1.07 (1.01-1.13) | | 5.2 | 7.92×10⁻⁶ |
| Urate | 1.07 (1.02-1.14) | 1.07 (1.01-1.13) | | 8.3 | 1.91×10⁻⁵ |

**b**  Usually *vs.* Never/rarely to MASLD

| Mediators | HR (95% CI) without adjusting for mediator | HR (95% CI) adjusting for mediator | Proportion explained by mediator (%) | | *P* for mediation |
|---|---|---|---|---|---|
| CRP | 1.19 (1.13-1.26) | 1.19 (1.12-1.25) | | 3.5 | 9.49×10⁻⁸ |
| IGF–1 | 1.20 (1.13-1.27) | 1.16 (1.09-1.23) | | 18.8 | 1×10⁻³⁵⁰ |
| Triglycerides | 1.19 (1.13-1.26) | 1.18 (1.12-1.25) | | 4.5 | 1×10⁻³⁵⁰ |
| Urate | 1.19 (1.12-1.26) | 1.19 (1.12-1.26) | | 2.0 | 0.008 |

**c**  Always *vs.* Never/rarely to MASLD

| Mediators | HR (95% CI) without adjusting for mediator | HR (95% CI) adjusting for mediator | Proportion explained by mediator (%) | | *P* for mediation |
|---|---|---|---|---|---|
| ALP | 1.33 (1.26-1.41) | 1.32 (1.24-1.39) | | 3.5 | 7.94×10⁻¹⁰ |
| CRP | 1.33 (1.25-1.40) | 1.31 (1.24-1.39) | | 4.7 | 1×10⁻³⁵⁰ |
| GGT | 1.33 (1.26-1.41) | 1.27 (1.19-1.34) | | 17.4 | 1×10⁻³⁵⁰ |
| HbA1c | 1.33 (1.26-1.41) | 1.31 (1.24-1.39) | | 4.8 | 1×10⁻³⁵⁰ |
| IGF-1 | 1.33 (1.26-1.41) | 1.27 (1.20-1.34) | | 17.5 | 1×10⁻³⁵⁰ |
| Triglycerides | 1.33 (1.26-1.41) | 1.32 (1.25-1.40) | | 2.8 | 1×10⁻³⁵⁰ |
| Urate | 1.33 (1.26-1.41) | 1.32 (1.25-1.40) | | 2.3 | 4.18×10⁻⁶ |

**Fig. 2 | The longitudinal mediation effect of 30 blood biomarkers on the association between the frequency of adding salt to foods and MASLD. a** Associations between sometimes *vs.* never/rarely adding salt to foods and the MASLD risk. **b** Associations between usually *vs.* never/rarely adding salt to foods and the MASLD risk. **c** Associations between always *vs.* never/rarely adding salt to foods and the MASLD risk. All results were adjusted for age, sex, ethnicity, Townsend deprivation index, education level, healthy diet score, smoking status, alcohol consumption, physical activity level, metabolic syndrome severity, cancer, and cardiovascular disease. Blue, purple, and red bars represent proportions explained by different mediators, the numbers to the right of each bar represent the proportion of the total effect explained by each mediator. Precise sample sizes for each biomarker are provided in Supplementary Table 3. Source data for Fig. 2 can be found in Supplementary Data 2. Abbreviation: *ALP* alkaline phosphatase, *CI* confidence interval, *CRP* C-reactive protein, *GGT* gamma-glutamyltransferase, *HR* hazard ratio, *IGF-1* insulin-like growth factor-1, *MASLD* metabolic dysfunction-associated steatotic liver disease, *SHBG* sex hormone-binding globulin.

Notably, we found that the association between the frequency of adding salt to foods and MASLD risk was stronger among individuals with a normal BMI, compared to those with overweight or obesity. In individuals with overweight or obesity, the adverse effects of excess adiposity may overshadow or mask the impact of salt intake, which was also observed in a previous study[17]. Our results also indicated a more pronounced association between the frequency of adding salt to foods and MASLD risk among regular alcohol consumers (three or more times per week) compared to non-drinkers. It suggests that individuals who frequently consume alcohol may be particularly susceptible to the adverse effects of high salt intake on MASLD, in consistent with the previous animal experimental study[48]. Similar interaction pattern was observed in an epidemiological study, in which the interactive association was found between salt intake, alcohol consumption and hypertension[49].

Our analysis of FIB-4 scores among individuals with MASLD showed that participants who always added salt to foods had over three-fold increased risk of advanced liver fibrosis compared to those who never added

salt to food. These findings add important clinical relevance to our study, suggesting that excessive salt intake may not only contribute to the development of MASLD but also to its progression, and highlight the potential value of dietary salt reduction in mitigating disease severity among patients with MASLD. However, it is crucial to note that FIB-4 is a surrogate marker and has limitations in accurately diagnosing advanced fibrosis across various clinical settings[50–55]. Future studies using more definitive measures of liver fibrosis, such as liver biopsy or advanced imaging techniques, are needed to confirm these associations.

The observed association between adding salt to foods and MASLD risk can be explained by several potential mechanisms. First, high salt intake has been shown to promote insulin resistance, a key feature in the pathogenesis of MASLD[56]. Our mediation analysis supports this pathway, as we found that HbA1c and IGF-1 partially mediated the association between the frequency of adding salt to foods and MASLD risk. Second, high salt intake may lead to increased oxidative stress and inflammation in the liver, as evidenced by experimental animal models[57]. This is consistent with our

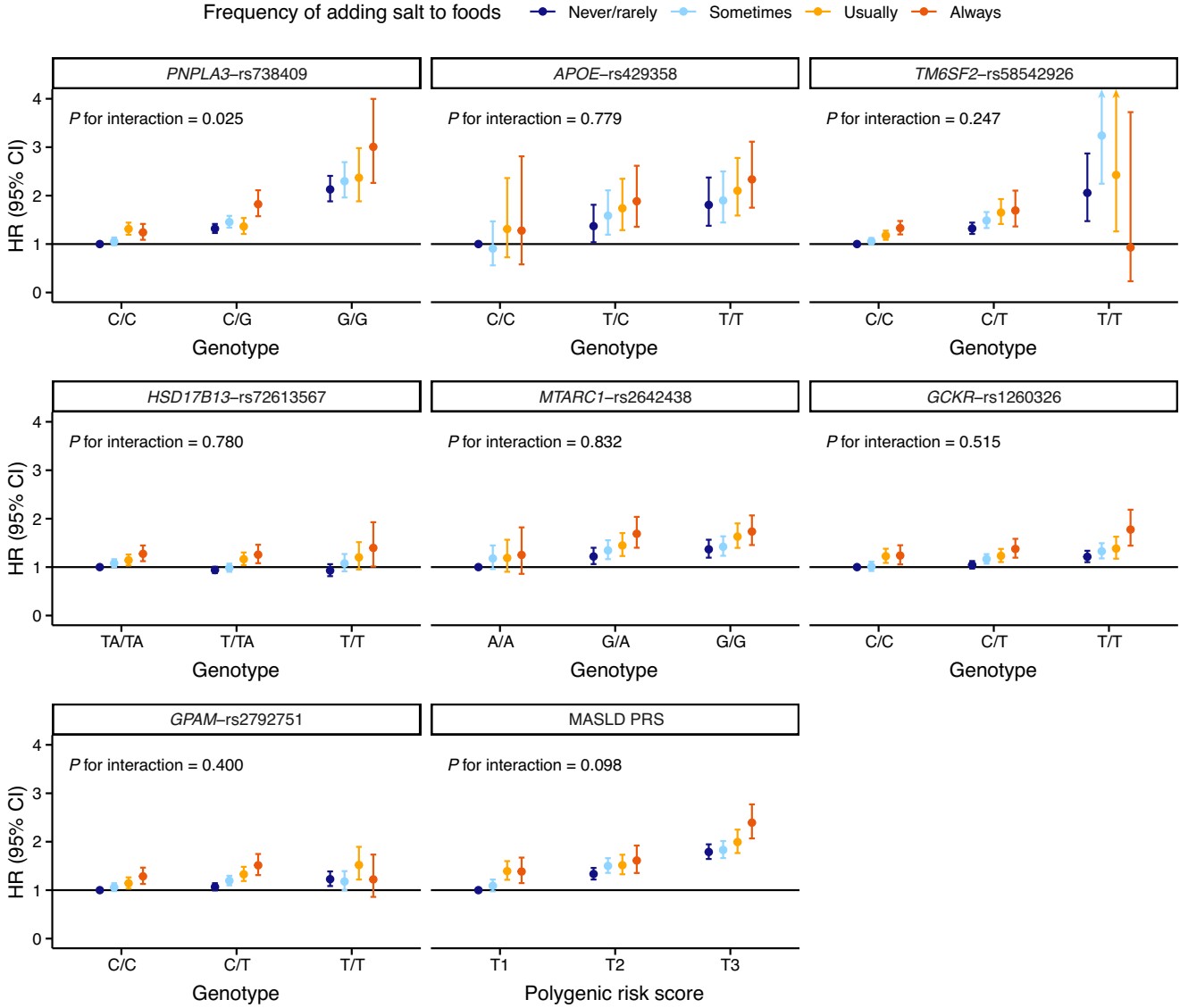

**Fig. 3 | The joint interaction effect of genetic predictors and the frequency of adding salt to foods on MASLD.** The figure shows HRs with 95% CIs for MASLD risk derived from multivariable-adjusted Cox regression models ($n$ = 479,589). The $x$-axis shows genotypes for individual SNPs or tertiles (T1-T3) for MASLD PRS. The $y$-axis shows HRs (point) with 95% CI (error bars). From left to right, dark blue circles with lines for never/rarely, light blue circles with lines for sometimes, orange circles with lines for usually, and dark orange circles with lines for always adding salt to foods. All results were adjusted for age, sex, ethnicity, Townsend deprivation index, education level, healthy diet score, smoking status, alcohol consumption, physical activity level, metabolic syndrome severity, cancer, and cardiovascular disease. Source data for Fig. 3 can be found in Supplementary Data 3. Abbreviation: *CI* confidence interval, *HR* hazard ratio, *MASLD* metabolic dysfunction-associated steatotic liver disease, *PRS* polygenic risk score.

finding that CRP, a marker of systemic inflammation, partially mediated the association between frequency of adding salt to foods and MASLD risk. Third, high salt intake has been associated with alterations in lipid metabolism[58], which aligns with our observation that triglycerides played a mediating role in the relationship between the frequency of adding salt to foods and MASLD risk.

Our study also revealed a significant interaction between the frequency of adding salt to foods and the *PNPLA3*-rs738409 gene variant in relation to MASLD risk. *PNPLA3* encodes a protein known as adiponutrin, involving in hepatic lipid metabolism[59]. *PNPLA3*-rs738409 is a well-established genetic risk factor for MASLD. Carriers of the risk allele (G) exhibit increased hepatic fat accumulation and impaired triglyceride hydrolysis, potentially exacerbating the effects of dietary salt intake on liver health[60]. In this study, we observed that participants who always added salt to foods and carried two risk alleles (G/G genotype) had a 3-fold higher risk of MASLD compared to those who never/rarely added salt to food and carried no risk

alleles. This underscores the potential for targeted dietary interventions in genetically susceptible individuals to significantly reduce MASLD risk.

Additionally, we found a nominal interaction between the frequency of adding salt to foods and the MASLD PRS. Both these interactions highlight the complex interplay between dietary factors and genetic predispositions in the development of MASLD. Although we did not observe statistically significant interactions for the remaining variants, individuals with a high frequency of adding salt to foods and carrying two risk alleles for these variants generally had a higher risk of MASLD compared to those with a low frequency of adding salt to foods and carrying two non-risk alleles.

Our study has several strengths. First, the large sample size and prospective design allow for a robust assessment of the temporal relationship between the frequency of adding salt to foods and MASLD risk. Second, we used multiple measures, including self-reported frequency of adding salt to foods, estimated 24-hour urinary sodium excretion, and liver fat content measured by MRI, providing a comprehensive evaluation of both the

exposure and outcome. Third, we conducted extensive analyses to explore potential mediators and gene-environment interactions, offering valuable insights into the underlying mechanisms.

However, some limitations should be considered. First, the frequency of adding salt to foods was self-reported and assessed only at baseline, which may not capture changes in this behavior over time. Nevertheless, we observed good consistency across multiple assessments of frequency of adding salt to foods (Supplementary Table 9). Second, the high frequency of adding salt to foods might be confounded by unhealthy lifestyle and dietary patterns. However, we have carefully adjusted for lifestyle factors, and both the subgroup analyses and sensitivity analyses indicated that the positive association of the frequency of adding salt to foods with risk of MASLD remained consistent. Third, our study utilized ICD-10 codes for MASLD identification, which predate the recent transition from NAFLD to MASLD terminology. MASLD require the presence of at least one cardiometabolic risk factor whereas NAFLD do not. However, we examined the prevalence of cardiometabolic risk factors among participants with NAFLD at baseline in this cohort. Among 583 individuals with baseline NAFLD, 99.3% ($n = 579$) had at least one cardiometabolic risk factor, suggesting that nearly all would meet the current MASLD criteria. This high concordance suggests that the use of the NAFLD definition is unlikely to have substantially affected our findings. Nonetheless, future studies applying the updated MASLD criteria are warranted. Forth, it is important to note that the frequency of adding salt to foods, while serving as a behavioral marker of salt preference, represents only one component of total dietary sodium intake[61]. Therefore, while our findings suggest that reducing the frequency of adding salt to foods may help lower MASLD risk, comprehensive sodium reduction strategies should focus on total dietary salt intake from all sources. Finally, the UK Biobank cohort may not be fully representative of the general population, potentially limiting the generalizability of our findings. Further randomized clinical trials are warranted to verify our results.

Our study demonstrates that a higher frequency of adding salt to foods is significantly associated with an increased risk of MASLD, with metabolic and inflammatory biomarkers, such as IGF-1, CRP, triglycerides, and urate partially mediating this relationship. Additionally, a significant interaction between frequency of adding salt to foods and the *PNPLA3*-rs738409 genetic variant was identified, underscoring the importance of reducing salt intake, particularly among genetically predisposed individuals, as a potential strategy for MASLD prevention and management.

## Data availability

The main data used in this study are accessed from the UK Biobank Resource (https://www.ukbiobank.ac.uk) under application number 95817. Due to data privacy regulations, these data cannot be shared by the authors directly. The UK Biobank data are available to approved researchers through application via the UK Biobank website. The source data for Fig. 2 is in Supplementary Data 2. The source data for Fig. 3 is in Supplementary Data 3.

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

## Acknowledgements

This work was supported by grants from the Basic Public Welfare Research Program of Zhejiang Province (Q24H260023), the Natural Science Foundation of Fujian Province (2023J05199), the Scientific Research Foundation of Hangzhou Normal University (4265C50223204007), and the High Level Introduction of Talent Research Start-up Fund of Putian University (2023020). The funder had no role in the design of the study, collection, analysis, and interpretation of data. This study was conducted using the UK Biobank resource (application 95817). We want to express our sincere thanks to the participants of the UK Biobank, and the members of the survey, development, and management teams of this project. Figure 1 was created using https://BioRender.com.

## Author contributions

Q.W. and S.L. were involved in the conception, design, and conduct of the study and the analysis and interpretation of the results. H.C. wrote the first draft of the manuscript, and all authors edited, reviewed, and approved the final version of the manuscript. H.C. analyzed the data and interpretated the results. X.Z., Q.W., and S.L. have made critical revision of the manuscript for important intellectual content. S.L. had full access to all the data in the study, and takes responsibility for the integrity of the data and the accuracy of the data analysis.

## Competing interests

The authors declare no competing interests.
