## [Transparent Peer Review file · Communications Medicine]

Adding salt to foods increases the risk of metabolic dysfunction-associated steatotic liver disease

Corresponding Author: Dr Qiong Wu

Version 0:

Reviewer comments:

Reviewer #1

(Remarks to the Author)

This is an absolutely interesting and well-rounded article. Excellent methodological approach, presentation and discussion of your results. Congratulations to the authors.

My only concern is that the new term "MASLD" has different diagnosis criteria compared to the old "NAFLD" term and herein the old criteria have been used. I recommend adding a relevant comment in the text to explain this discrepancy, although the right way to do it is to re-run the analysis with the correct participants.

I have no further comments to add.

PS: In line 119 please add "s" in "participant"

Reviewer #2

(Remarks to the Author)

In their study using data from the UK Biobank, the authors examined the relationship between the frequency of adding salt to foods and the development of MASLD. They found a positive correlation, indicating that a higher frequency of adding salt was associated with an increased risk of MASLD. The study conveys a key message that is not only scientifically relevant but also practical and easily translatable to daily lifestyle choices. I have only some minor comments before considering for publication.

1. The following definition is NAFLD: "It is characterized by excessive fat accumulation in the liver in the absence of other causes, such as viral or autoimmune hepatitis, medication, drugs, and excessive alcohol intake.". MASLD (Metabolic Dysfunction–Associated Steatotic Liver Disease) is characterized by excess fat accumulation in the liver accompanied by at least one criterion of metabolic dysfunction (J Hepatol. 2023 Dec;79(6):1542-1556.). Please correct this issue.
2. I recommend referencing this manuscript, which also investigated the association between salt intake and MASLD using data from the UK Biobank: Nutr Res. 2025 Apr;136:94-104.
3. Do the authors have the opportunity to assess MASLD severity, for example by calculating the FIB-4 index? FIB-4 is proposed to be used in MASLD in first line stratification of the disease severity (Aliment Pharmacol Ther. 2020 Aug;52(3):513-526). Including such an analysis would allow evaluation of the association between salt addition and the severity of MASLD, which could significantly enhance the impact and clinical relevance of the manuscript.
4. A FIB-4 score greater than 2.67 is commonly used to indicate a high risk of advanced fibrosis. However, if the authors are able to calculate FIB-4, it is important to acknowledge that this non-invasive test is primarily validated for ruling out advanced fibrosis rather than for its definitive diagnosis. This limitation has been demonstrated across various clinical settings. I recommend addressing this point in the discussion as a methodological limitation (J Clin Gastroenterol. 2020 Nov/Dec;54(10):891-897.; Acta Diabetol. 2020 May;57(5):613-618; Postgrad Med. 2022 May;134(4):435-440.; Hepatology Forum 2020; 1: 8–13; Gut Liver 2020; 14: 486–491; Eur J Gastroenterol Hepatol. 2022 Jan 1;34(1):98-103.).
5. In another prospective study, increased salt intake was primarily associated with a lack of awareness. The consumption of convenience foods and frequent eating outside the home contributed significantly to overall salt exposure, particularly among males. In that cohort, the practice of adding extra salt during meals was relatively uncommon. Therefore, it is not only the act of adding salt at the table, but the total amount of salt consumed that is most relevant to MASLD development. While reducing the frequency of adding salt may help lower risk, the focus should be on total dietary salt intake. I recommend

discussing this aspect to provide a more comprehensive understanding of the relationship between salt consumption and MASLD: Nutrients. 2023 Sep 12;15(18):3942.

Version 1:

Reviewer comments:

Reviewer #1

(Remarks to the Author)

Thank you for addressing all comments. I agree with your addition in the discussion section but I would expect an addition in the methodology section regarding the criteria used for patients identification which are actually based on NAFLD diagnosis and this should be explained in the methods.

Reviewer #2

(Remarks to the Author)

The authors improved their work significantly. I have no further comments. In this current form, the study is suitable for publication.

Responses to Reviewer #1:

Comment 1: This is an absolutely interesting and well-rounded article. Excellent methodological approach, presentation and discussion of your results. Congratulations to the authors.

Response: Many thanks for your positive comments and constructive feedback. We sincerely appreciate your thorough review of our manuscript.

Comment 2: My only concern is that the new term "MASLD" has different diagnosis criteria compared to the old "NAFLD" term and herein the old criteria have been used. I recommend adding a relevant comment in the text to explain this discrepancy, although the right way to do it is to re-run the analysis with the correct participants.

Response: Thank you very much for your insightful comment. We fully acknowledge that the recently proposed term "MASLD" differs conceptually from the old term "NAFLD" with the MASLD criteria requiring at least one abnormal cardiometabolic risk factor. The shift from NAFLD to MASLD terminology highlights the importance of considering metabolic dysfunction. We also agree that reanalysis using the updated MASLD definition would represent a more accurate approach.

However, due to limitations in the UK Biobank follow-up data, comprehensive cardiometabolic risk indicators were not available for all participants, making it infeasible to fully apply the MASLD criteria in our current analysis. In response to your suggestion, we have added a relevant statement in the revised Discussion section to clarify this discrepancy and to acknowledge it as a limitation of the study.

To further address the potential impact, we try to examined the prevalence of cardiometabolic risk factors among participants with NAFLD at baseline. We found that 99.3% (n = 579 out of 583) had at least one abnormal cardiometabolic risk factor, indicating that nearly all would meet the MASLD definition. This finding is consistent with a recent study by Younossi et al. (J Hepatol. 2024 May;80(5):694-701), which reported that 99.8% of NAFLD patients fulfilled MASLD criteria

(Cohen's kappa = 0.968), suggesting strong concordance between the two definitions. Therefore, we believe the use of the NAFLD definition is unlikely to have substantially influenced our findings.

The clarification has been added to the revised manuscript, as shown below:

Discussion section, Page 11, Line 296-305: *“Third, our study utilized ICD-10 codes for MASLD identification, which predate the recent transition from NAFLD to MASLD terminology. MASLD require the presence of at least one cardiometabolic risk factor whereas NAFLD do not. However, we examined the prevalence of cardiometabolic risk factors among participants with NAFLD at baseline in this cohort. Among 583 individuals with baseline NAFLD, 99.3% (n = 579) had at least one cardiometabolic risk factor, suggesting that nearly all would meet the current MASLD criteria. This high concordance suggests that the use of the NAFLD definition is unlikely to have substantially affected our findings. Nonetheless, future studies applying the updated MASLD criteria are warranted.”*

Comment 3: I have no further comments to add.

Response: We sincerely thank you for your positive assessment of our work.

Comment 4: PS: In line 119 please add "s" in "participant"

Response: We apologize for this oversight and have corrected “participant” to “participants” in line 119. We have also carefully reviewed the entire manuscript for any similar typographical errors.

Responses to Reviewer #2:

Comment 1: In their study using data from the UK Biobank, the authors examined the relationship between the frequency of adding salt to foods and the development of MASLD. They found a positive correlation, indicating that a higher frequency of adding salt was associated with an increased risk of MASLD. The study conveys a key message that is not only scientifically relevant but also practical and easily

translatable to daily lifestyle choices. I have only some minor comments before considering for publication.

Response: We greatly appreciate your positive comments and constructive feedback. We have thoroughly addressed each query below to enhance the clarity and robustness of our findings.

Comment 2: The following definition is NAFLD: “It is characterized by excessive fat accumulation in the liver in the absence of other causes, such as viral or autoimmune hepatitis, medication, drugs, and excessive alcohol intake.”. MASLD (Metabolic Dysfunction–Associated Steatotic Liver Disease) is characterized by excess fat accumulation in the liver accompanied by at least one criterion of metabolic dysfunction(J Hepatol. 2023 Dec;79(6):1542-1556.). Please correct this issue.

Response: Thank you very much for this important correction. We have revised the definition of MASLD in the Introduction section and added the appropriate reference (J Hepatol. 2023 Dec;79(6):1542-1556). The sentence has been corrected in the revised manuscript, as follows:

Introduction section, Page 1, Line 20-23: *“Metabolic dysfunction-associated steatotic liver disease (MASLD) has recently been proposed to replace the previously used term non-alcoholic fatty liver disease. It is characterized by excessive fat accumulation in the liver accompanied by at least one criterion of metabolic dysfunction.¹”*

Comment 3: I recommend referencing this manuscript, which also investigated the association between salt intake and MASLD using data from the UK Biobank: Nutr Res. 2025 Apr:136:94-104.

Response: Thank you for bringing this relevant study to our attention. We have reviewed the referenced article by Liu Q et al. (*“High salt diet causally increases metabolic dysfunction-associated steatotic liver disease risk: A bidirectional Mendelian randomization study”*), which provided robust genetic evidence that “salt added to food” is a causal risk factor for MASLD by using multiple genome-wide

association study datasets. This study offers valuable context and reinforces the importance of dietary salt intake in MASLD research. Accordingly, we have cited and briefly discussed this work in the revised Introduction and Discussion section, as shown below:

Introduction section, Page 3, Line 51-56: *“Recently, a Mendelian randomization study provided genetic evidence supporting a potential causal relationship between genetically-proxied salt preference and MASLD risk. While these findings enhance the biological plausibility of the salt–MASLD link, prospective population-based studies assessing dietary behaviors remain limited and are needed to validate this association in real-world settings “.*

Discussion section, Page 9, Line 221-226: *“In addition, a Mendelian randomization study using multiple genome-wide association study datasets identified “salt added to food” as a potential causal risk factor for MASLD, providing genetic support for our findings.²¹ Building upon this, our study contributes complementary observational evidence by evaluating salt-adding behavior in a large prospective cohort, accounting for multiple potential confounders and conducting a series of secondary analyses.”*

Comment 4: Do the authors have the opportunity to assess MASLD severity, for example by calculating the FIB-4 index? FIB-4 is proposed to be used in MASLD in first line stratification of the disease severity (Aliment Pharmacol Ther. 2020 Aug;52(3):513-526). Including such an analysis would allow evaluation of the association between salt addition and the severity of MASLD, which could significantly enhance the impact and clinical relevance of the manuscript.

Response: Thank you very much for your excellent suggestion. In response, we assessed MASLD severity at baseline by calculating the Fibrosis-4 (FIB-4) index based on age, aspartate aminotransferase (AST), alanine aminotransferase (ALT), and platelet count among participants with MASLD (n = 524). The FIB-4 index was estimated as follows (age in years, ALT and AST in IU/L, and platelet count in 10⁹/L):

$$\text{FIB-4 index} = \frac{(\text{age} \times \text{AST})}{\text{platelet count} \times \sqrt{\text{ALT}}}$$

The mean (SD) for FIB-score was 1.66 (1.05) and the proportion of individuals classified as high-risk for advanced fibrosis (FIB-4 > 2.67) was 49 (10.3%).

We further analyzed the association between the frequency of adding salt to foods and both continuous and binary FIB-4 outcomes in individuals with MASLD at baseline. As shown in the **Table 1** below and in the revised **Supplementary Table 5**. For continuous FIB-4 scores, individuals who always added salt to foods showed higher FIB-4 values compared to those who never/rarely added salt ($\beta = 0.29$, 95% CI: -0.05 to 0.63 in Model 2), although this association did not reach statistical significance. However, when FIB-4 was analyzed as a binary outcome, those who always added salt to foods had significantly higher odds of advanced fibrosis (OR = 3.24, 95% CI: 1.19-8.36) compared to the reference group. No significant associations were observed in the “sometimes” and “usually” adding salt to food groups.

These findings add important clinical relevance to our study by demonstrating that higher salt intake is associated not only with MASLD incidence but also with disease severity. We have incorporated this analysis into the Methods, Results, and Discussion sections of the revised manuscript, as follows:

Methods section, Page 14, Line 377-384: *“We calculated the FIB-4 index based on age, aspartate aminotransferase (AST), alanine aminotransferase (ALT), and platelet count among participants with MASLD at baseline.²⁷ The FIB-4 index was estimated as follows (age in years, ALT and AST in IU/L, and platelet count in 10⁹/L). Following established criteria, we classified participants with FIB-4 > 2.67 as having a high risk of advanced fibrosis, and those with FIB-4 ≤ 2.67 as having a low/indeterminate risk of liver fibrosis.^{52,53”}*

Methods section, Page 16, Line 443-445: *“The associations between the frequency of adding salt to foods and continuous FIB-4 index and binary FIB-4 index was evaluated using multivariable linear regression and logistic regression, respectively.”*

Results section, Page 5, Line 118-129: *“To evaluate the association between the*

frequency of adding salt to foods and MASLD severity, we calculated the Fibrosis-4 (FIB-4) index among participants with MASLD at baseline.²⁷ Among 524 participants with MASLD at baseline who had complete data for FIB-4 index calculation, we examined the association between the frequency of adding salt to foods and liver fibrosis risk. In the fully adjusted model, compared to those who never/rarely added salt to foods, participants who always added salt showed a trend toward higher FIB-4 scores ($\beta = 0.29$, 95% CI: -0.05 to 0.63), although this did not reach statistical significance for the continuous outcome. However, when analyzed as a binary outcome, participants who always added salt to foods demonstrated a significantly higher risk of advanced fibrosis (FIB-4 > 2.67) compared to the reference group (OR = 3.24, 95% CI: 1.19-8.36) (Supplementary Table 5).”

Discussion section, Page 9, Line 239-245: “Our analysis of FIB-4 scores among individuals with MASLD showed that participants who always added salt to foods had over three-fold increased risk of advanced liver fibrosis compared to those who never added salt to food. These findings add important clinical relevance to our study, suggesting that excessive salt intake may not only contribute to the development of MASLD but also to its progression, and highlight the potential value of dietary salt reduction in mitigating disease severity among patients with MASLD.”

Table 1. Association between frequency of adding salt to foods and FIB-4 index in individuals with MASLD at baseline.

	Frequency of adding salt to foods, β (95% CI)				P for trend
	Never/rarely	Sometimes	Usually	Always	
FIB-4 (continuous)					
No. of participants	251	147	84	42	
Model 1	0 (reference)	0.15 (-0.06, 0.35)	-0.07 (-0.32, 0.19)	0.39 (0.06, 0.72)	0.15
Model 2	0 (reference)	0.14 (-0.08, 0.35)	-0.10 (-0.35, 0.16)	0.29 (-0.05, 0.63)	0.39
	Frequency of adding salt to foods, OR (95% CI)				P for trend
	Never/rarely	Sometimes	Usually	Always	
FIB-4 (binary)					
Events No./total No.	19/251	13/147	8/84	9/42	
Model 1	1 (reference)	1.21 (0.56-2.53)	1.11 (0.44-2.60)	3.75 (1.47-9.08)	0.03
Model 2	1 (reference)	1.15 (0.52-2.49)	1.01 (0.38-2.45)	3.24 (1.19-8.36)	0.08

We classified participants with FIB-4 index > 2.67 as having a high risk of advanced fibrosis, and those with FIB-4 \leq 2.67 as having a low/indeterminate risk of liver fibrosis.

Model 1: Adjusted for age, sex

Model 2: Model 1 + ethnicity, Townsend deprivation index, education level, healthy diet score, smoking status, alcohol consumption, physical activity level, metabolic syndrome severity, cancer, and cardiovascular disease

Comment 5: A FIB-4 score greater than 2.67 is commonly used to indicate a high risk of advanced fibrosis. However, if the authors are able to calculate FIB-4, it is important to acknowledge that this non-invasive test is primarily validated for ruling out advanced fibrosis rather than for its definitive diagnosis. This limitation has been demonstrated across various clinical settings. I recommend addressing this point in the discussion as a methodological limitation (J Clin Gastroenterol. 2020 Nov/Dec;54(10):891-897.; Acta Diabetol. 2020 May;57(5):613-618; Postgrad Med. 2022 May;134(4):435-440.; Hepatology Forum 2020; 1: 8–13; Gut Liver 2020; 14: 486–491; Eur J Gastroenterol Hepatol. 2022 Jan 1;34(1):98-103.).

Response: We appreciate your important clarification regarding the FIB-4 interpretation. To address this, a statement acknowledging this limitation has been added to the Discussion section. The recommended references have also been cited to support the statement. The revised manuscript shows as follows:

Discussion section, Page 9-10, Line 245-249: *“However, it is crucial to note that FIB-4 is a surrogate marker and has limitations in accurately diagnosing advanced fibrosis across various clinical settings.^{32–37} Future studies using more definitive measures of liver fibrosis, such as liver biopsy or advanced imaging techniques, are needed to confirm these associations.”*

Comment 6: In another prospective study, increased salt intake was primarily associated with a lack of awareness. The consumption of convenience foods and frequent eating outside the home contributed significantly to overall salt exposure, particularly among males. In that cohort, the practice of adding extra salt during meals was relatively uncommon. Therefore, it is not only the act of adding salt at the table, but the total amount of salt consumed that is most relevant to MASLD development. While reducing the frequency of adding salt may help lower risk, the focus should be on total dietary salt intake. I recommend discussing this aspect to provide a more comprehensive understanding of the relationship between salt consumption and

MASLD: Nutrients. 2023 Sep 12;15(18):3942.

Response: Thank you very much for your insightful comment and for recommending this relevant reference. We fully agree that the frequency of adding salt to food reflects only one aspect of overall dietary salt intake (such as salt preference) and does not reflect total sodium intake. Other salt sources such as processed foods, restaurant meals, or low salt awareness, as indicated in the cited study, also significantly have impact on daily salt intake. Future research that incorporates more comprehensive measures of salt intake are needed. We have incorporated the suggested reference and added this limitation, as follows:

Discussion section, Page 11-12, Line 305-310: *“Forth, it is important to note that the frequency of adding salt to foods, while serving as a behavioral marker of salt preference, represents only one component of total dietary sodium intake.⁴³ Therefore, while our findings suggest that reducing the frequency of adding salt to foods may help lower MASLD risk, comprehensive sodium reduction strategies should focus on total dietary salt intake from all sources.”*

Responses to Reviewer #1:

Comment 1: Thank you for addressing all comments. I agree with your addition in the discussion section but I would expect an addition in the methodology section regarding the criteria used for patients' identification which are actually based on NAFLD diagnosis and this should be explained in the methods.

Response: Thank you for this important clarification. We have now revised the *Outcome Assessment section* in the Methods to clearly state that patient identification was based on diagnostic codes for non-alcoholic fatty liver disease (NAFLD). This addition enhances the transparency of our case identification approach. The revised text is shown below:

Methods section, Page 7-8, Line 160-166: *“It should be noted that while we use the term MASLD throughout this manuscript to align with current nomenclature, case identification was based on the previous NAFLD criteria, as the ICD-10 coding system predates the recent transition from NAFLD to MASLD terminology. According to the Expert Panel Consensus Statement ⁴⁶, incident MASLD was defined using International Classification of Diseases-10th Revision (ICD-10) codes K76.0 for nonalcoholic fatty liver disease and K75.8 for nonalcoholic steatohepatitis.”*

Responses to Reviewer #2:

Comment 1: The authors improved their work significantly. I have no further comments. In this current form, the study is suitable for publication.

Response: We sincerely thank you for your positive assessment and for your valuable feedback throughout the review process, which has significantly strengthened our manuscript.